# Immune Contexture of MMR-Proficient Primary Colorectal Cancer and Matched Liver and Lung Metastases

**DOI:** 10.3390/cancers13071530

**Published:** 2021-03-26

**Authors:** Maarit Ahtiainen, Hanna Elomaa, Juha P. Väyrynen, Erkki-Ville Wirta, Teijo Kuopio, Olli Helminen, Toni T. Seppälä, Ilmo Kellokumpu, Jukka-Pekka Mecklin

**Affiliations:** 1Department of Education and Research, Hospital Nova of Central Finland, 40620 Jyväskylä, Finland; Teijo.Kuopio@ksshp.fi (T.K.); jukka-pekka.mecklin@ksshp.fi (J.-P.M.); 2Department of Biological and Environmental Science, University of Jyväskylä, 40014 Jyväskylä, Finland; hanna.elomaa@ksshp.fi; 3Department of Pathology, Hospital Nova of Central Finland, 40620 Jyväskylä, Finland; juha.vayrynen@ksshp.fi; 4Cancer and Translational Medicine Research Unit, Medical Research Center Oulu, Oulu University Hospital, and University of Oulu, 90014 Oulu, Finland; 5Department of Gastroenterology and Alimentary Tract Surgery, Tampere University Hospital, 33520 Tampere, Finland; Erkki-Ville.Wirta@pshp.fi; 6Surgery Research Unit, Medical Research Center Oulu, Oulu University Hospital, and University of Oulu, 90014 Oulu, Finland; olli.helminen@oulu.fi; 7Department of Surgery, Hospital Nova of Central Finland, 40620 Jyväskylä, Finland; ilmo.kellokumpu@fimnet.fi; 8Department of Surgery, Helsinki University Central Hospital, and University of Helsinki, 00014 Helsin-ki, Finland; 9Surgical Oncology, Johns Hopkins University, Baltimore, MD 21218, USA; 10iCAN Digital Precision Cancer Medicine Flagship, Helsinki University Central Hospital, and University of Helsinki, 00014 Helsinki, Finland; 11Faculty of Sport and Health Sciences, University of Jyväskylä, 40014 Jyväskylä, Finland

**Keywords:** colorectal cancer, metastases, tumour infiltrating lymphocytes, PD-1, PD-L1

## Abstract

**Simple Summary:**

Metastasis is the main cause for cancer mortality. The most common metastatic sites of colorectal cancer (CRC) are the liver and lungs. Tumour-infiltrating lymphocytes are recognized as beneficial prognostic factors both in primary and metastatic CRC, but less is known about their reciprocal differences. The aim of our study was to evaluate immune microenvironment and its prognostic value in a series of mismatch proficient (pMMR) CRC with matched liver and lung metastases. The proportion of tumours with high immune cell infiltration together with PD-L1-positivity almost doubled in metastases compared to primary tumours. Our study confirmed the prognostic value of high ICS in least immune-infiltrated metastases in pMMR CRC patients. Major differences observed in immune contexture between primary tumours and metastases may have significance for treatment strategies for patients with advanced CRC.

**Abstract:**

Purpose: To evaluate immune cell infiltration, the programmed death-1/programmed death ligand-1 (PD-1/PD-L1) expression and their prognostic value in a series of mismatch proficient (pMMR) CRC with matched liver and lung metastases. Methods: Formalin-fixed paraffin-embedded tissue sections stained for CD3, CD8, PD-L1 and PD-1 from 113 primary CRC tumours with 105 liver and 59 lung metastases were analyzed. The amount of CD3 and CD8 positive lymphocytes were combined as immune cell score (ICS). Comparative analyses on immune contexture were performed both between the primary tumour and matched metastases and between the metastatic sites. Results: In liver metastases, immune cell infiltration was increased in general compared to primary tumours but did not correlate case by case. On the contrary, ICS between lung metastases and primary tumours correlated well, but the expression of PD-1/PD-L1 was increased in lung metastases. The proportion of tumours with high ICS together with PD-L1-positivity almost doubled in metastases (39%) compared to primary tumours (20%). High ICS (compared to lowest) in patient’s least immune-infiltrated metastasis was an independent prognostic marker for disease-specific (HR 9.14, 95%CI 2.81–29.68) and overall survival (HR 6.95, 95%CI 2.30–21.00). Conclusions: Our study confirms the prognostic value of high ICS in least immune-infiltrated metastases in pMMR CRC patients. Major differences observed in immune contexture between primary tumours and metastases may have significance for treatment strategies for patients with advanced CRC.

## 1. Introduction

Colorectal cancer (CRC) is the third most common cancer worldwide and second in terms of cancer mortality [1,2]. The most common metastatic sites of CRC are the liver and lungs [3,4]. Approximately 15–25% of CRC patients have liver metastases at the time of primary diagnosis [5,6] and around equal number of patients will develop metastases within the next 5 years [7]. Lung metastases are found in about 10% of patients with newly diagnosed CRC and in 5% of patients within the next 5 years [8]. Five-year overall survival (OS) for CRC patients without metastasis at the diagnosis of the primary tumour is 75–90%, whereas for patients with synchronous or metachronous metastases, the mean OS remains under 20% [9,10,11]. As the prognosis of patients with advanced CRC is dramatically impaired, more effective therapeutic strategies are needed. Solitary and even multiple metastases are increasingly within the limits of curative surgical treatment [6].

Immune escape is a notable hallmark of cancer [12]. Various subsets of immune cells are identified in tumour microenvironment, and they can either prevent tumour development or promote tumour progression and metastasis [13]. The abundance of tumour-infiltrating lymphocytes (TILs) has been shown to have significant prognostic value both in local and advanced colorectal cancer [14,15,16,17,18]. Several studies have shown discordance in immune cell infiltration between primary tumour and metastases, but these studies have focused mainly on liver metastases, only [19,20]. The most frequent metastatic sites of CRC, liver and lung, are immunologically very different. Lung is an organ with a highly active local immune system (reviewed by [21]), whereas liver is considered as an immunosuppressive organ [22]. How these immunological differences impact on response to recently generalized immune modulating therapies, is not currently known.

About 15% of colorectal cancers are characterized by mismatch repair (MMR) deficiency, leading to the generation of immunogenic neopeptides that enhance the anti-tumour immune response. A promising response to immune checkpoint inhibitors has been seen in MMR-deficient solid tumours, regardless of their primary site [23,24], leading to them being approved by FDA, as a cancer group, an indication for immune checkpoint inhibitor treatment. However, the MMR-deficiency is infrequent in advanced CRC [25]. Nevertheless, an intensified immune infiltration has been associated with better survival also in metastatic CRC [26,27], but metastases and primary tumours have also been shown to differ regarding their immune infiltration [19,20].

Since the immune therapies are currently indicated only to be used for advanced CRC rather than non-metastatic CRC, the knowledge of immune contexture in metastases and whether it differs from the primary tumour may be essential for patient’s treatment. The present study aimed to clarify this question by analysing the surgically resected liver and lung metastases in a population-based, consecutive and thoroughly characterized series of CRCs. We performed a quantification of immune cells and immune checkpoint (PD-1/PD-L1) expression in primary CRC tumour samples together with matched liver and lung metastases. Whole section slides were thoroughly analysed, and the results were related to patient’s clinical characteristics and survival.

## 2. Results

### 2.1. Patient Demographics

A total of 1671 patients met the criteria of having been diagnosed and treated for CRC in the Central Finland Hospital District during the study period. Metastatic disease was diagnosed in 551 (33.0%), of which 296 (17.7%) had synchronous metastases. Of 1302 resected Stage I-III patients, 255 (19.6%) developed metachronous metastases. The overall metastasectomy rate was 16.2% with synchronous metastases and 23.9% with metachronous metastases [6]. Our study material included 113 CRC patients with operable liver and/or lung metastases operated at the Central Finland Central Hospital (CFCH) in Jyväskylä during the years 2000–2018. A total of 72 CRC patients with liver metastasis, 23 patients with lung metastasis and 18 patients having both liver and lung metastasis were included in this study. Since some of the patients had several metachronous metastases, the total number of analysed liver and lung metastases was 105 and 59, respectively.

The clinicopathological characteristics of the patients and the characteristics of metastases are presented in Table 1 and Table 2, respectively. The median age of patients at diagnosis of the primary tumour was 66 years and the age at onset was similar between the patients with liver and lung metastasis (*p* = 0.253). The stage of CRC at the time of the diagnosis was higher, whereas the tumour grade was lower in patients with liver metastases (*p* < 0.001). There was no difference in the location of the primary tumour between colon and rectum in selection to metastasectomy. All patients undergoing resection for lung metastases only had recurrent disease (metachronous lung metastases) (*p* < 0.001). Preoperative chemotherapy as well as histological tumour response were more frequent with patients with liver metastases (*p* = 0.001 and 0.008, respectively). All primary tumours with resected liver or lung metastases within the 19-year study period were mismatch repair proficient (pMMR).

### 2.2. Immune Contexture in Primary Colorectal Cancer and Matched Liver and Lung Metastases

Figure 1 shows the representative IHC staining for CD3, CD8, PD-1 and PD-L1. The densities of CD3, CD8 and PD-1 lymphocytes in primary tumours and matched metastases (liver-only, lung-only and both) were analysed both in TC and IM and are shown in Figure 2. The density of TILs was higher in IM than in TC both in primary tumours and in metastases in all groups (Figure 2). CD3 and CD8 densities in IM were significantly higher in liver metastases compared to primary tumours (*p* < 0.001). This was seen also in patients having both liver and lung metastases (*p* < 0.001). Instead, the density of TILs did not differ between lung metastases and corresponding primary tumours. Compared to lung metastases, CD8 density in IM was significantly higher in liver metastases (*p* = 0.005, Table 2).

The density of PD-1 positive cells was significantly higher in IM of lung metastases compared to corresponding primary tumours (*p* = 0.034, Figure 2). Compared to liver metastases, PD-1 density in TC and IM was significantly higher in lung metastases (*p* = 0.001 and 0.003, respectively, Table 2). Categorized immune variables in primary tumours and matched metastases are shown in Appendix A. PD-L1 expression was mainly found on immune cells rather than on tumour cells both in primary tumours and in metastases. PD-L1 positive tumour cells were only found in one primary tumour, in two liver metastases and in three lung metastases. By contrast, PD-L1 expression on immune cells (IC), principally determining the PD-L1 positivity, was commonly seen both in primary tumours and in metastases. PD-L1 expression was significantly higher in lung metastases compared to both primary tumours (*p* < 0.001, Appendix A) and liver metastases (*p* = 0.001, Table 2). When the immune cell densities were compared according to the time of appearance of metastases, less CD3 and CD8 lymphocytes in TC (*p* = 0.005 and <0.001, respectively) were found in metachronous compared to synchronous metastases. No differences in PD-1 density or PD-L1 positivity were seen between these groups. 

We also analysed correlations between immune infiltrates in primary tumours and matched metastases. In patients with lung metastases, CD3 in TC (*r* = 0.365, *p* = 0.006) and in IM (*r* = 0.395, *p* = 0.004) as well as CD8 in TC (*r* = 0.349, *p* = 0.008) and in IM (*r* = 0.455, *p* = 0.001) all moderately correlated with CD8 density of IM in corresponding primary tumours. In addition, PD-1 in TC and IM in lung metastases moderately correlated with PD-1 in TC of primary tumours (*r* = 0.314, *p* = 0.018 and r = 0.404, *p* = 0.003, respectively). Also, PD-L1 in IC correlated between lung metastases and primary tumours (*r* = 0.338, *p* = 0.011). In patients with liver metastases, neither TILs nor PD-1/PD-L1 correlated between primary tumours and metastases (Figure 3). 

ICS was more often high in liver metastases compared to primary tumours (*p* = 0.025, Appendix A). Immunoprofiles differed significantly both in liver and lung metastases compared to their primary tumours (*p* = 0.045 and 0.009, respectively, Appendix A). Altogether, ICS strongly associated with the density of PD-1 positive cells and PD-L1 positivity both in primary tumours and metastases. When individual immune parameters were combined into a Tumour Immunity in the MicroEnvironment (TIME) classification (Figure 4), remarkable differences between primary tumours and metastases were seen (*p* < 0.001). The proportion of high immune cell infiltration (TIME2 and TIME3) was higher in metastases (49.7%) compared to primary tumours (33.4%) (*p* = 0.008). Furthermore, the proportion of cancers with presence of PD-L1 together with high immune cell infiltration (TIME2) almost doubled in metastases (39.4%) compared to primary tumours (20.4%) (*p* = 0.008). TIME subtypes either in primary tumours or in metastases did not associate with DSS or OS. 

Metastatic immune densities were also studied in relation with the response to preoperative treatment. ICS was more frequently high in metastases with histological tumour response (TRGs 1–3) compared with the metastasis having no response (TRGs 4–5) (87% vs. 63%, *p* = 0.003). Differences were not seen in PD-1 or PD-L1 positivity between response-based groups. 

### 2.3. Prognostic Impact of Immune Contexture in Primary Tumours and Metastases

Since primary tumour resection, the mean survival time of patients with operated liver, lung and both metastases were 5.0 ± 3.7, 6.4 ± 4.1 and 5.4 ± 2.4 years, respectively (*p* = 0.362). Time between primary tumour resection and the first metachronous metastasis was 1.1 ± 1.2, 2.5 ± 1.4 and 1.0 ± 1.1 years, respectively (*p* < 0.001). There were no cases with operated synchronous lung metastases. CRC was the cause of death in 57% of patients. Three- and 5-year DSS rates after metastasectomy were 60% and 51%, 48% and 30%, and 72% and 56% for patients with liver, lung and both metastases, respectively. Accordingly, 3- and 5-year OS rates after metastasectomy were 49% and 39%, 39% and 22% and 72% and 44%, respectively (Table 1).

Figure 5 shows Kaplan-Meier curves for the entire cohort (OS and DSS, Figure 5A) and Forest Plots showing subgroup results (Figure 5B). Among the analysed clinicopathological variables, high TNM stage of primary disease was prognostic for poor survival, with stage IV having 40% 5-year DSS and 34% 5-year OS (*p* = 0.011 and 0.006, respectively). The survival of patients with metachronous metastases was better compared to patients with synchronous metastases, with 62% 5-year DSS and 55% 5-year OS (*p* = 0.021 and 0.020, respectively). Among immune variables, high ICS of the patient’s least-infiltrated metastasis (Figure 6) was significantly prognostic for better survival outcome, with 5-year DSS of 17% for ICS0 versus 63% for ICS4 as well as 5-year OS of 17% for ICS0 versus 55% for ICS4 (*p* = 0.001 and 0.002, respectively). 

According to the univariable analysis, stage of primary disease, primary tumour grade, onset of metastases, primary tumour location, size of metastases, ICS and PD-L1^IC^ were included in the multivariable analysis together with age and sex. Table 3 shows the multivariable model with separate immune parameters for ICS and PD-L1^IC^. ICS of patient’s least-infiltrated metastasis was found to be an independent prognostic marker. ICS0, with respect to ICS4 as a reference, had a DSS hazard ratio (HR) of 9.14 and an OS HR of 6.95 (*p* < 0.001 and 0.001, respectively). High stage of primary disease was prognostic for worse OS (HS 9.44, *p* = 0.045) and high grade of primary tumour for worse DSS (HR 4.18, *p* = 0.035). Furthermore, rectal tumour location had a worse DSS (HR 2.19, *p* = 0.008). Combining ICS, PD-1 and PD-L1 as immunoprofile in metastases did not enhance the prognostic performance. 

## 3. Discussion

We studied a reasonable number of liver and lung metastases together with primary tumours from mismatch repair proficient (pMMR) patients to expand the knowledge of immune microenvironment in these tumours. Immune cell infiltration in liver metastases was increased but did not correlate with TIL density of the corresponding primary tumours. In lung metastases, the number of TILs moderately correlated with the density in primary tumours, and PD-1 and PD-L1 expressions were increased in lung metastases compared to primary tumours. Furthermore, the proportion of tumours with high immune cell infiltration together with PD-L1-positivity almost doubled in metastases compared to primary tumours. In this study cohort, high ICS in patient’s least-infiltrated metastasis was proven to have prognostic value in pMMR CRC patients.

Multiple data support the major role of immune infiltrates within primary CRC tumours in predicting the survival of patients [16,17,18,28]. Increasing evidence also exist on the positive impact of intrametastatic immune infiltrates on patients with advanced CRC [27,29,30] as well as their response to chemotherapy [26]. Our results confirm the previous observation that ICS in the least immune-infiltrated metastases has prognostic value [27,30]. However, the density of PD-1 positive cells and PD-L1 positivity did not increase the prognostic value of ICS, which was previously seen with primary CRC and small bowel adenocarcinomas [31,32]. Still, these immune factors were strongly associated with each other also in metastases. 

Few publications have reported a comparison of immune microenvironment between CRC metastases and corresponding primary tumours. Previous studies have shown that liver metastases of CRC differ from primary tumours [15,19,30]. Metastases of the same patient have also diverse amounts of immune cells [27,30,33]. Lung metastases in CRC are less studied with discordant results [34,35]. Our study shows a few interesting differences between lung and liver metastases. Equally to previous studies [27,30,35], but in contrast to the study by Remark et al. [34], the density of TILs was higher in both liver and lung metastases compared to primary tumours. Interestingly, immune cell infiltration in liver metastases was increased in general compared to primary tumours, but not in case by case and PD-1/PD-L1 expression was increased only in lung metastases. It is plausible that the less adapted tumour microenvironment of the metastatic sites recruits more host immune cells attracted by the unidentified cancer cells than that of the primary tumour site. Recently, specific changes in cancer-related genes and immune cell infiltration in CRC metastases have been related to metastatic evolution, which produce new perspectives for cancer diagnostics and therapeutic strategies [36,37].

Immunotherapy has been demonstrated to benefit some patients with mismatch repair deficient (dMMR) CRC [38,39]. Presumably, this results from high mutation burden due to the defective DNA mismatch repair and the following high neoantigen density, which primes T-cells to strong antitumour immune responses [40,41]. TIME classification has been proposed as a frameshift for tailoring cancer immunotherapy [42,43]. Recently, the associations between TIME subtypes and clinical, pathological, and molecular characteristics of CRC were studied [44]. Among other characteristics, TIL-present subtypes (TIME 2 and 3); the proposed candidates for immunotherapy, were associated with dMMR. However, stage 4 dMMR tumours constitute only 4% of all advanced CRCs [45]. For the vast majority of advanced CRC that are pMMR, alternative treatment approaches are needed. In our study cohort, there was no case of metastatic dMMR CRC which demonstrates the difference in the behaviour between dMMR and pMMR CRC. dMMR CRC seems to be less capable to develop distant metastases. To our knowledge, this is the first study showing the distribution of TIME subgroups between primary CRC tumours and metastases. Our results show that the proportion of tumours potentially responsive to immunotherapy is highly increased in metastases compared to primary tumours in pMMR CRC patients. Essential changes in immune pattern seem to occur during the progress of CRC that should not be ignored, when tailoring the treatment for patients with advanced CRC. However, combination strategies are apparently needed for improving treatment efficacy in pMMR CRC [46].

Our study has some limitations. The number of patients with resected metastases is relatively small and inclusion of only resectable patients may subject to selection bias. However, the studied patients belong to a well-characterized population-based cohort of patients with CRC (n = 1671) in Central Finland in 2000–2015 with comprehensive follow-up and reliable cancer recurrence and survival data [6]. Strength of our study is the evaluation of metastatic immune infiltration from whole tumour sections. As shown previously [30], biopsy or a TMA core of a metastasis rarely represents the whole lesion, due to the intra- and inter-tumour heterogeneity of immune infiltrates. The evaluation of PD-L1 expression from biopsies has been shown even more unreliable [30]. Overall, our results highlight the importance of studing the immune contexture also in metastases in addition to primary tumour in advanced CRC.

## 4. Materials and Methods

### 4.1. Tumour Sampling

All patients diagnosed with primary CRC during years 2000–2016 were identified using the histopathological registry of the Central Finland Central hospital, which covers all CRCs diagnosed in the Central Finland area (population catchment area of approximately 280,000). All patients with metastasectomy of liver or lung were included and only operated metastases are involved in the tables.

In cases of a metachronous CRC or a local recurrence, the analyses were performed on the patient´s first CRC sample. The metastases were classified synchronous, if diagnosed at the same time or before the primary tumour, and metachronous, if diagnosed in the postoperative surveillance. Cancer progression and survival of the patients were followed until the censoring date end of June 2020. Causes of death were updated in June 2020 from the Finnish Cause of Death Registry. The examination of the primary CRC and metastasis tissue was performed by a pathologist following the AJCC guidelines (8th edition). Tumour samples were graded (grades 1–4) based on the percentage of glandular formation according to the World Health Organization (WHO) criteria [44].

### 4.2. Immunohistochemical Analyses

Both the 113 primary CRC tumours with 105 liver and 59 lung metastases were included in analogous immunohistochemical analyses. Formalin-fixed paraffin-embedded whole tissue sections of 3 μm thickness were used. Mismatch repair status was determined by immunohistochemical analysis for the expression of MLH1, PMS2, MSH2, and MSH6 as described previously [47]. Staining for PD-1 (The Human Genome Organization (HUGO) name PDCD1) and PD-L1 (HUGO name CD274) was conducted with anti-PDCD1 (SP269, 1:50; Spring Bioscience, Pleasanton, CA, USA) and anti-CD274 (E1L3N, 1:100; Cell Signaling Technology, Danvers, MA, USA) antibodies, using a BOND-III stainer (Leica Biosystems, Buffalo Grove, IL, USA). Staining for CD3 and CD8 was conducted with anti-CD3 (LN 10, 1:200; Leica Biosystems, Newcastle, UK) and anti-CD8 (SP16, 1:400; Thermo Scientific, Fremont, CA, USA) antibodies, using a Lab Vision Autostainer 480 (ImmunoVision Technologies Inc., Brisbane, CA, USA). Signal visualization was done by diaminobenzidine and sections were counterstained with haematoxylin. Slides were scanned with a NanoZoomer-XR (Hamamatsu Photonics, Hertfordshire, UK) at ×20 magnification.

### 4.3. Scoring

Scoring was conducted as described earlier [31], and both primary tumours and metastases were analysed similarly. Briefly, positively stained CD8, CD3 and PD-1 lymphocytes (per 1 mm^2^) were calculated from the representative areas of tumour centre and invasion margin by using QuPath [48]. The invasive margin was selected manually using an annotation brush with a diameter of 720 μm [49]. The mean analysed area from CD3, CD8- and PD-1-stained sections was 36 mm^2^. Cut-off values for ICS were selected from receiver operating characteristic curves (815 for CD3+ and 384 for CD8+ in the tumour centre (TC) and 1144 for CD3+ and 496 for CD8+ in the invasive front (IM) of primary tumours (Rajamäki K. (University of Helsinki, Helsinki, Finland). Personal communication, 2020), and 94 for CD3+ and 33 for CD8+ in the TC and 1322 for CD3+ and 664 for CD8+ in the IM of metastases) in relation to disease-specific 3-year mortality. Accordingly, cut-off value for PD-1 positivity was selected from receiver operating characteristic curve (15 positive cells per mm^2^ in primary tumours and 27 in metastases). 

ICS was formulated following the example of Galon et al. [28], as presented in our previous studies [18,31]. Patients were divided into low ICS (scores 0–2) and high ICS (scores 3–4) groups for further analysis. The impact of immune cell densities on survival was analysed from the metastasis with the smallest number of immune cells called least-infiltrated metastasis. Tumour samples were also categorized into four different TIME-classes [42] based on the presence of TILs (ICS low/high) and PD-L1 expression (PD-L1 positivity/negativity).

PD-L1 expression was evaluated on tumour cells (TC) and tumour infiltrating immune cells (IC) throughout the TC and IM as described previously [31]. Both the percentage of stained tumour cells and immune cells and the staining intensity were visually estimated. Tumour sample was defined as PD-L1 positive when ≥5% of the tumour cells and/or tumour infiltrating immune cells was positive for PD-L1 with moderate or strong intensity.

Tumour regression was scored for each metastasis according to the scheme of Rubbia-Brandt [50], where the tumour regression grades (TRGs) 1–5 are based on the presence of residual tumour cells and the extent of fibrosis. Five TRGs were further categorized into three groups: major or complete histological tumour response (MjHR; TRG1 and TRG2), partial histological tumour response (PHR; TRG 3) and no histological tumour response (NHR; TRG 4 and TRG5).

### 4.4. Statistical Analysis

Categorical data were compared using the Pearson’s chi-square test. The Kaplan-Meier method was used to calculate disease-specific survival (DSS) and overall survival (OS), and the log-rank test was used to compare differences. Survival times for DSS and OS were calculated from the date of primary surgery to the date of death or the end of follow-up. Death within 30 days following surgery was considered postoperative. Univariable and multivariable Cox proportional hazards regression models were used to analyse prognostic factors for DSS and OS. Only variables with a *p* value of < 0.20 in univariable analysis were included in the multivariable analysis with age and sex. Statistical analysis was performed using IBM SPSS Statistics (version 23.0; SPSS Inc., Chicago, IL, USA).

### 4.5. Ethical Aspects 

The study was approved by the ethical committee of the Central Finland Central Hospital and the National Supervisory Authority for Welfare and Health (Valvira, Helsinki, Finland).

## 5. Conclusions

In conclusion, our study shows differences in immune contexture between primary tumours and matched metastases in advanced CRC. The proportion of tumours with high immune cell infiltration together with PD-L1-positivity almost doubled in metastases compared to primary tumours. Our study also confirms that high ICS in patient’s least-infiltrated metastasis has prognostic value in pMMR CRC patients.

Major differences observed in the immune environment of the primary tumour and metastatic sites reflect the immune-avoiding capabilities acquired by the migrant tumour cell population and the tumour-benefiting circumstances of tumour microenvironment allowing the metastatic tumorigenesis. These differences may be somewhat stochastic, but common denominators of the immune cell densities were revealed by our well-validated quantification tools. The characteristics and trends identified in the current study may help in designing anti-cancer therapy schemas that make the most out of the host immune response performance and enable further work in identifying which patients in the pMMR subcohorts might benefit from immunomodulative treatment options.

## Figures and Tables

**Figure 1 cancers-13-01530-f001:**
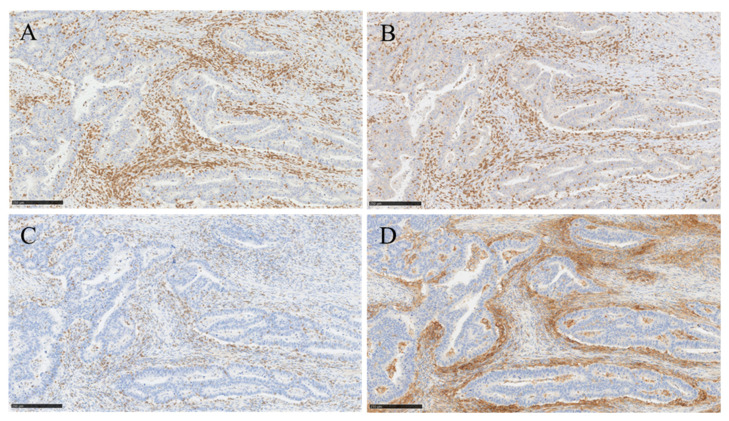
Immunohistohistochemical staining of CD3, CD8, PD-1 and PD-L1 in primary colorectal tumour. (**A**,**B**) show representative immunostaining of high infiltrates of CD3 (**A**) and CD8 lymphocytes (**B**), respectively. (**C**,**D**) show representative immunostaining of high PD-1 expression (**C**) and strong expression of PD-L1 in immune cells (**D**) surrounding the tumour area (10× magnification, scale bar 250 µm).

**Figure 2 cancers-13-01530-f002:**
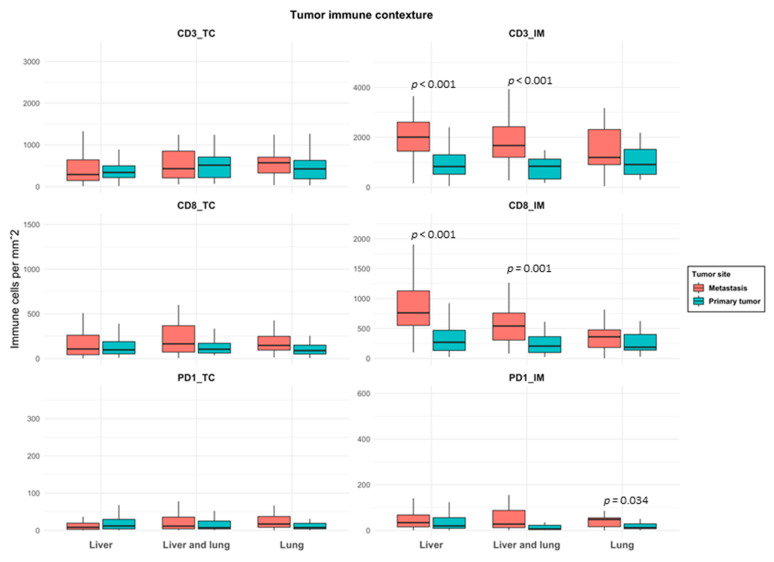
Box plot of immune cell densities in primary tumours and matched metastases in patients with liver-only, liver and lung and lung-only metastases. Horizontal line indicates the median, box the interquartile range (IQR) and whiskers the smallest/largest value no further than 1.5 × IQR. Abbreviations: TC: tumour center; IM: invasive margin.

**Figure 3 cancers-13-01530-f003:**
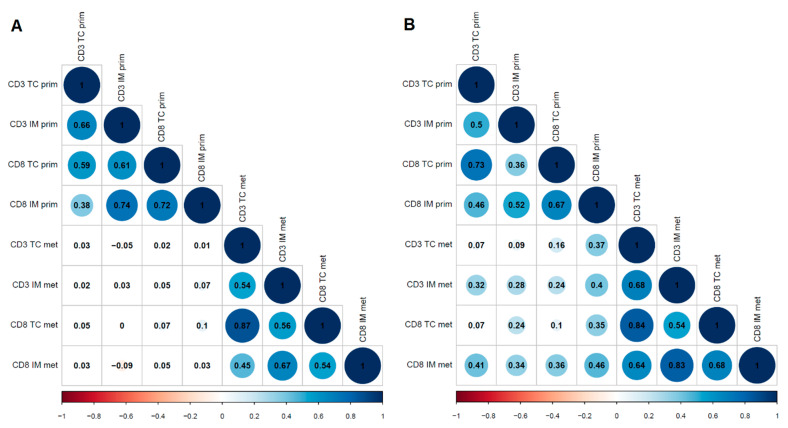
Correlograms of T cell densities in colorectal adenocarcinomas and their metastases ((**A**): Liver; (**B**): Lung). The numbers indicate Spearman’s rank correlation coefficients. Abbreviations: IM: invasive margin; prim: primary tumour; met: metastasis; TC: tumour center.

**Figure 4 cancers-13-01530-f004:**
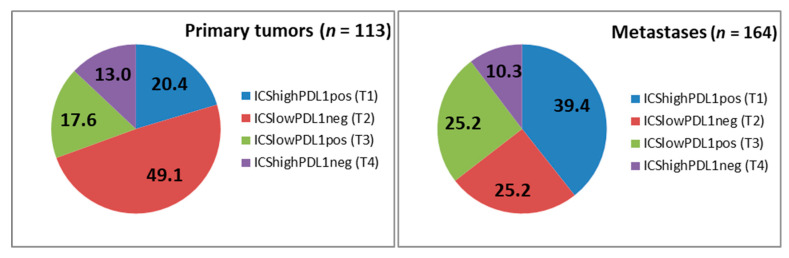
TIME classification of primary tumours and metastases. Subtypes are immunological ignorance; T1 (ICSlowPDL1neg), adaptive immune resistance; T2 (ICShighPDL1pos), tolerance; T3 (ICShighPDL1neg) and intrinsic induction; T4 (ICSlowPDL1pos). The values represent the percentage of each subtype. (ICS = Immune Cell Score).

**Figure 5 cancers-13-01530-f005:**
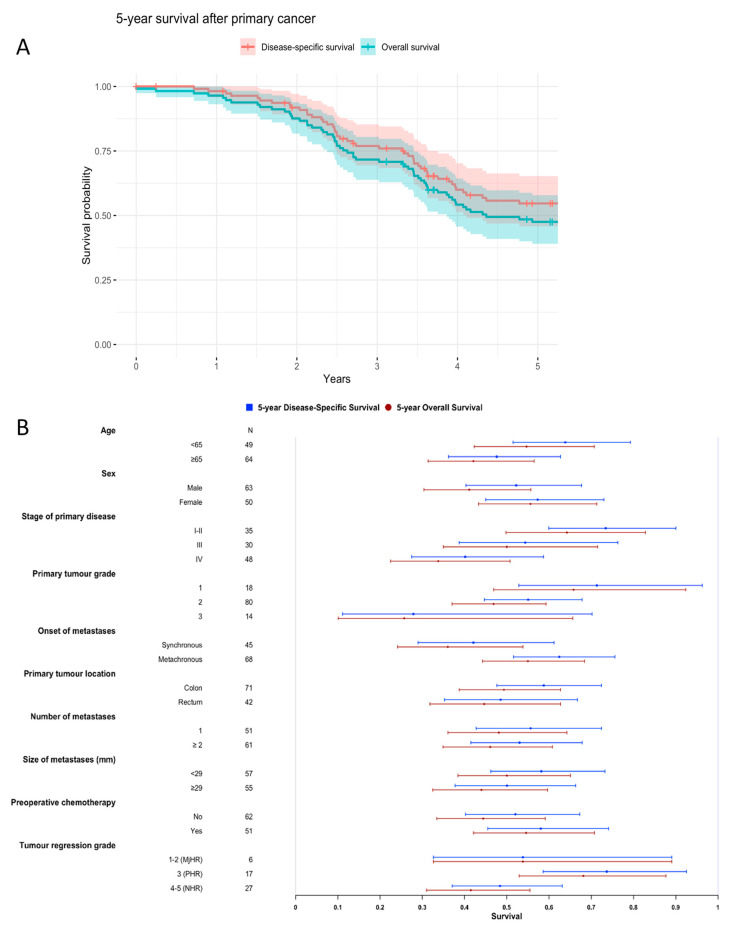
Kaplan-Meier curves for the entire cohort (OS and DSS) (**A**) and Forest plots for subgroup results (**B**).

**Figure 6 cancers-13-01530-f006:**
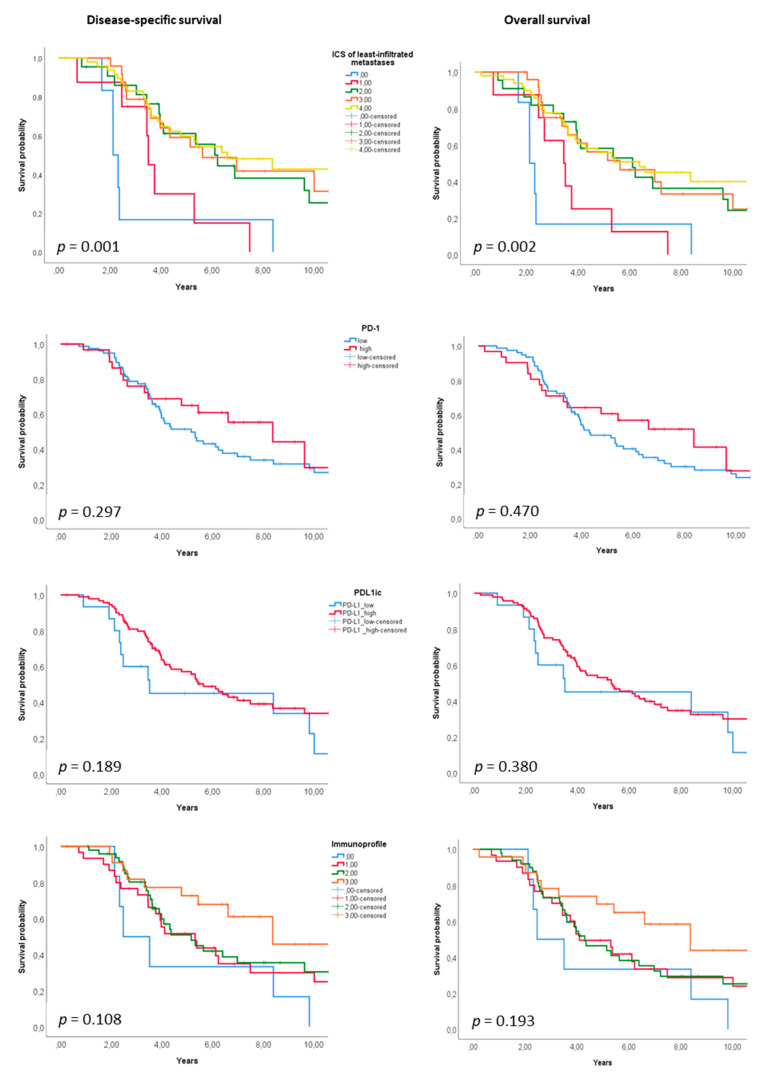
Disease-specific and overall survival according to ICS 0–4, PD-1, PD-L1IC and immunoprofile of least-infiltrated metastases.

**Table 1 cancers-13-01530-t001:** Cinicopathological variables in patients with metastatic colorectal cancer.

Characteristics	Location of Metastases		
	Liver (% of Column)	Lung (% of Column)	Liver and Lung (% of Column)
Total No of Patients	72	23	18
Age			
<65	58 (81)	15 (65)	15 (83)
≥65	14 (19)	8 (35)	3 (17)
Gender			
Male	43 (60)	10 (44)	10 (56)
Female	29 (40)	13 (56)	8 (44)
Stage of disease			
I	3 (4)	0	3 (17)
II	16 (22)	11 (48)	3 (17)
III	14 (19)	12 (52)	4 (22)
IV	39 (54)	0	8 (44)
Primary tumour grade			
1	7 (10)	2 (9)	9 (50)
2	56 (79)	16 (69)	8 (44)
3	8 (11)	5 (22)	1 (6)
Timing of metastases			
Synchronous	37 (51)	0	8 (44)
Metachronous	35 (49)	23	10 (56)
Primary tumour location			
Colon	51 (71)	11 (48)	11 (61)
Rectum	21 (29)	12 (52)	7 (39)
Metastases			
Mean no./patient (min-max)	2.1 (1–12)	1.6 (1–4)	3.4 (2–7)
Mean size (mm)/patient (min-max)	34 (7–105)	27 (7–90)	37 (15–70)
Preoperative chemotherapy			
No	32 (44)	21 (91)	9 (50)
Yes	40 (56)	2 (9)	9 (50)
Survival after metastasectomy			
3-year DSS	43 (60)	11 (48)	13 (72)
5-year DSS	37 (51)	7 (30)	10 (56)
3-year OS	35 (49)	9 (39)	13 (72)
5-year OS	28 (39)	5 (22)	8 (44)

Abbreviations: DSS, disease specific survival; OS, overall survival. Primary tumour grade is missing from one patient with liver metastasis.

**Table 2 cancers-13-01530-t002:** Metastases’ characteristics.

Total Number of Metastases	Liver Metastases (% of Column)	Lung Metastases (% of Column)	*p*-Value
	105	59	
Primary tumor location			
colon	73 (70)	28 (55)	0.005
rectum	32 (30)	31 (45)	
Mean size (mm)	34.5	31.4	0.148
Timing of metastases			
Synchronous	56 (53)	11 (19)	<0.001
Metachronous	49 (47)	48 (81)	
Preoperative chemotherapy			
No	45 (43)	44 (75)	<0.001
Yes	60 (57)	15 (25)	
Tumour regression grade			
1–2 (MjHR)	8 (13)	1 (7)	
3 (PHR)	25 (42)	3 (20)	0.146
4–5 (NHR)	27 (45)	11 (73)	
TILs density *			
CD3 ^TC^	504 ± 537	651 ± 535	0.334
CD3 ^IM^	2098 ± 877	1596 ± 926	0.269
CD8 ^TC^	219 ± 272	223 ± 248	0.232
CD8 ^IM^	894 ± 483	438 ± 374	0.005
PD-1 ^TC^	19 ± 34	41 ± 67	0.001
PD-1 ^IM^	53 ± 62	79 ± 116	0.003
Immune cell score			
0	18 (18)	12 (22)	0.230
1	13 (13)	7 (13)	
2	21 (21)	19 (34)	
3	13 (13)	6 (11)	
4	35 (35)	11 (20)	
PD-L1 ^TC^			
neg	102 (98)	56 (95)	0.261
pos	2 (2)	3 (5)	
PD-L1 ^IC^			
neg	49 (47)	12 (20)	0.001
pos	55 (53)	47 (80)	

Abbreviations: MjHR, major or complete histological tumour response; PHR, partial histological tumour response, NHR, no histological tumour response; TILs, tumour infiltrating lymphocytes; PD-1, programmed cell death protein 1; PD-L1, programmed death ligand 1; TC, tumour cell; IM, invasive margin; IC, immune cell; *, cells per 1 mm^2^. ICS was indeterminable from five liver and four lung metastases and PD-L1 from one liver metastasis.

**Table 3 cancers-13-01530-t003:** Multivariable analysis with Cox proportional hazard model.

Characteristics	Univariable Analysis (DSS)		Univariable Analysis (OS)		Multivariable Analysis (DSS)		Multivariable Analysis (OS)	
	HR (95% CI)	*p*-Value	HR (95% CI)	*p*-Value	HR (95% CI)	*p*-Value	HR (95% CI)	*p*-Value
Age								
<65	1	0.503	1	0.234	1	0.294	1	0.088
≥65	1.18 (0.72–1.94)		1.33 (0.83–2.12)		1.38 (0.75–2.54)		1.64 (0.93–2.88)	
Sex								
Male	1	0.711	1	0.184	1	0.736	1	0.553
Female	0.91 (0.56–1.49)		0.73 (0.46–1.16)		0.90 (0.48–1.67)		0.84 (0.47–1.49)	
Stage of primary disease								
I-II	1		1		1		1	
III	1.31 (0.68–2.52)	0.036	1.36 (0.74–2.52)	0.018	0.96 (0.44–2.09)	0.500	1.25 (0.60–2.59)	0.045
IV	2.14 (1.17–3.91)		2.24 (1.26–3.95)		3.64 (0.37–35.7)		9.44 (1.60–55.82)	
Primary tumour grade								
1	1	0.103	1	0.109	1	0.035	1	0.132
2	1.62 (0.76–3.44)		1.65 (0.82–3.35)		1.68 (0.69–4.11)		1.46 (0.65–3.30)	
3	2.74 (1.08–6.97)		2.59 (1.07–6.27)		4.18 (1.36–12.80)		2.86 (1.01–8.11)	
Onset of metastases								
Synchronous	1.80 (1.09–2.97)	0.023	1.74 (1.08–2.81)	0.022	1.18 (0.13–10.77)	0.883	2.91 (0.54–15.67)	0.214
Metachronous	1		1		1		1	
Primary tumour location								
Colon	1	0.036	1	0.116	1	0.008	1	0.075
Rectum	1.69 (1.03–2.78)		1.46 (0.91–2.33)		2.19 (1.23–3.92)		1.66 (0.95–2.88)	
Size of metastases (mm)								
<29	1	0.103	1	0.188	1	0.340	1	0.455
≥29	1.52 (0.92–2.50)		1.37 (0.86–2.18)		1.36 (0.73–2.53)		1.25 (0.70–2.24)	
ICS of least-infiltrated metastases								
0	5.00 (2.01–12.42)	0.002	4.09 (1.67–9.99)	0.005	9.14 (2.81–29.68)	<0.001	6.95 (2.30–21.00)	0.001
1	2.90 (1.24–6.81)		2.72 (1.23–6.01)		5.23 (1.97–13.83)		4.76 (1.96–11.60)	
2	1.13 (0.58–2.19)		0.97 (0.52–1.83)		1.29 (0.57–2.88)		1.34 (0.62–2.88)	
3	1.20 (0.62–2.32)		1.14 (0.62–2.11)		1.17 (0.55–2.48)		1.31 (0.66–2.60)	
4	1				1		1	
PD-L1 ^IC^								
low	1.55 (0.80–2.97)	0.192	1.33 (0.70–2.55)	0.382	1.60 (0.67–3.79)	0.290	1.55 (0.68–3.50)	0.298
high	1		1		1		1	

Abbreviations: DSS, disease-specific survival; OS, overall survival HR, hazard ratio; CI, confidence interval; PD-1, programmed cell death protein 1; PD-L1, programmed death ligand 1; IC, immune cell. Analyses were performed with the following reference categories: <65 years, male gender, TNM Stage I-II, tumour grade 1, metachronous onset of metastases, tumour location in colon, size of metastases <29 mm, high Immune cell score, and high PD-L1IC. For analyses, 110 patients were available. One patient had unknown primary tumour grade, size of metastases was missing from one patient and PD-L1IC was indeterminable from two patients.

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
