# Peer review of "Immune Contexture of MMR-Proficient Primary Colorectal Cancer and Matched Liver and Lung Metastases"

_cancers, 2021, doi:10.3390/cancers13071530_

Round 1

Reviewer 1 Report

Tables, in general, are too many and too long.

Table 1 is purely descriptive, there should be no p-values given. And what test do the p-values refer to? lung vs. liver vs. liver and lung? Please omit.

Table 2: Images speak sounder than words. These results are perfect for presentation in the form of box, scatter or violin plots. If you absolutely want to show the table, move it the appendix.

Table 4 and 5. Again, I find the density of numbers confusing. Instead of Table 4, consider showing 2 Kaplan-Meier curves for the entire cohort (OS and DSS) plus Forest Plots showing subgroup results. Instead of Table 5, consider presenting 8 Kaplan Meier curves: ICS 0-4, PD-1, PD-L1 and immunoprofile. Omit ICS hi/lo.

Author Response

Response to Reviewer 1 Comments

Point 1: Tables, in general, are too many and too long.

Response 1: We agree with the reviewer that tables are too many and they are rather crowded. Improvements have been made as follows.

Point 2: Table 1 is purely descriptive, there should be no p-values given. And what test do the p-values refer to? lung vs. liver vs. liver and lung? Please omit.

Response 2: The column including p-values in Table 1 is removed.

Point 3: Table 2: Images speak sounder than words. These results are perfect for presentation in the form of box, scatter or violin plots. If you absolutely want to show the table, move it the appendix.

Response 3: The upper part of Table 2 (TILs densities) has been restructured as a box plot in Figure 2. The lower part of Table 2 including categorized variables has been moved into Supplementary Materials Table S1.

Point 4: Table 4 and 5. Again, I find the density of numbers confusing. Instead of Table 4, consider showing 2 Kaplan-Meier curves for the entire cohort (OS and DSS) plus Forest Plots showing subgroup results. Instead of Table 5, consider presenting 8 Kaplan Meier curves: ICS 0-4, PD-1, PD-L1 and immunoprofile. Omit ICS hi/lo.

Response 4: Table 4 is replaced with Figure 5 including Kaplan-Meier curves for the entire cohort (OS and DSS) (A) and Forest plots for subgroup results (B). Table 5 is replaced with Figure 6 including the suggested Kaplan-Meier curves. ICS hi/lo was omitted.

Reviewer 2 Report

Ahtiainen and colleagues aimed to characterize the immune contexture of MMR-proficient (pMMR) primary colorectal cancer (CRC) and matched liver and lung metastases. Toward this goal, formalin-fixed paraffin-embedded tissue sections from 113 primary CRC and 105 matched liver and 59 lung metastases were stained for protein expression of CD3, CD8, PD-1, and PD-L1.

Higher densities of CD3 and CD8 lymphocytes were observed at the invasive margin (IM) compared to the tumor center. Within the IM, CD3 and CD8 expression was significantly higher in liver metastases compared to primary tumors. When expression of CD3 and CD8 within lymphocytes was summarized as immune cell score (ICS), the respective ICS was often high in liver metastases compared to primary tumors, and the proportion of high immune cell infiltration was higher in metastases compared to primary tumors. Combining the individual immune parameters into a Tumor Immunity in the MicroEnvironment (TIME) classification revealed differences between metastases and primary tumors. Finally, the results of the protein expression analyses were related to patients´ clinical characteristics and survival. Here, high ICS of the patients’ least-infiltrated metastasis was significantly prognostic for better survival outcome.

While this study deals with a very interesting and clinically relevant topic, there are several critiques which hamper my enthusiasm:

  • To me, this study lacks clear novelty. Parts of these analyses have been published elsewhere (e.g. Kwak et al. Oncotarget. 2016 Dec 6;7(49):81778-81790; Liu et al. Sci Rep. 2020 Jul 1;10(1):10725; Wang et al. Clin Lab. 2020 Dec 1;66(12)). While it may be correct that other studies did not analyze matched pairs of primary tumors and metastases, I am not entirely convinced by the data.
  • It was really difficult to get through and understand the data and analyses. This manuscript consists of six huge tables, which should be reduced to only relevant information. Additional tables with less important information could be provided as supplementary material.
  • Figure 1: Please provide a scale bar for each panel.
  • In Table 3: “TILs density” is marked with an asterisk, which is not explained in the legend.
  • In the results part 2.3, the authors refer to Kaplan-Meier survival analyses. It would be helpful if some curves would be depicted as figure.
  • Page 8: In the last sentence, the authors refer to a Figure 4, which is missing in the manuscript.
  • Page 14: Section 4.2, please add an explanation for PDCD1 and CD274.

Author Response

Response to Reviewer 2 Comments

Ahtiainen and colleagues aimed to characterize the immune contexture of MMR-proficient (pMMR) primary colorectal cancer (CRC) and matched liver and lung metastases. Toward this goal, formalin-fixed paraffin-embedded tissue sections from 113 primary CRC and 105 matched liver and 59 lung metastases were stained for protein expression of CD3, CD8, PD-1, and PD-L1.

Higher densities of CD3 and CD8 lymphocytes were observed at the invasive margin (IM) compared to the tumor center. Within the IM, CD3 and CD8 expression was significantly higher in liver metastases compared to primary tumors. When expression of CD3 and CD8 within lymphocytes was summarized as immune cell score (ICS), the respective ICS was often high in liver metastases compared to primary tumors, and the proportion of high immune cell infiltration was higher in metastases compared to primary tumors. Combining the individual immune parameters into a Tumor Immunity in the MicroEnvironment (TIME) classification revealed differences between metastases and primary tumors. Finally, the results of the protein expression analyses were related to patients´ clinical characteristics and survival. Here, high ICS of the patients’ least-infiltrated metastasis was significantly prognostic for better survival outcome.

While this study deals with a very interesting and clinically relevant topic, there are several critiques which hamper my enthusiasm:

Point 1: To me, this study lacks clear novelty. Parts of these analyses have been published elsewhere (e.g. Kwak et al. Oncotarget. 2016 Dec 6;7(49):81778-81790; Liu et al. Sci Rep. 2020 Jul 1;10(1):10725; Wang et al. Clin Lab. 2020 Dec 1;66(12)). While it may be correct that other studies did not analyze matched pairs of primary tumors and metastases, I am not entirely convinced by the data.

Response 1: We agree that the subject of our study is not totally novel. However, only few publications have reported a comparison of immune microenvironment between CRC metastases and corresponding primary tumours. As our results show, remarkable differences exist which might have significance for treatment strategies for patients with advanced CRC.

Furthermore, the heterogeneity of immune infiltrates is well recognized. This increases the reliability of our results based on whole section analyses.

Point 2: It was really difficult to get through and understand the data and analyses. This manuscript consists of six huge tables, which should be reduced to only relevant information. Additional tables with less important information could be provided as supplementary material.

Response 2: We agree with the reviewer that tables are too many and they are rather crowded. Several improvements have been made as follows:

  • The upper part of Table 2 (TILs densities) has been restructured as a box plot in Figure 2. The lower part of Table 2 including categorized variables has been moved into Supplementary Materials Table S1.
  • Table 4 is replaced with Figure 5 including Kaplan-Meier curves for the entire cohort (OS and DSS) (A) and Forest plots for subgroup results (B).
  • Table 5 is replaced with Figure 6 including Kaplan-Meier curves.

Point 3: Figure 1: Please provide a scale bar for each panel.

Response 3: Scale bars have been added.

Point 4: In Table 3: “TILs density” is marked with an asterisk, which is not explained in the legend.

Response 4: We apologize for the missing explanation. This has been added.

Point 5: In the results part 2.3, the authors refer to Kaplan-Meier survival analyses. It would be helpful if some curves would be depicted as figure.

Response 5: Table 4 and Table 5 have been restructured as Figure 5 and Figure 6, respectively.

Point 6: Page 8: In the last sentence, the authors refer to a Figure 4, which is missing in the manuscript.

Response 6: We apologize for the missing Figure 4 due to a human error. As mentioned previously (Response 2), improvements with additional figures has been made. Accordingly, the references in text have been changed.

Point 7: Page 14: Section 4.2, please add an explanation for PDCD1 and CD274.

Response 7: PDCD1 and CD274 have been explained being the names according to HUGO in Section 4.2.

Reviewer 3 Report

TITLE:   Immune contexture of MMR-proficient primary colorectal cancer and matched liver and lung metastases.

Correspondence: Dr. Maarit Ahtiainen  (Department of Education and Research, Hospital Nova of Central Finland, 40620 Jyväskylä, Finland)

Comments to the Author

Summary of the major findings and impressions.

The aim of the study was to evaluate immune microenvironment and its prognostic value in a series of mismatch proficient (pMMR) CRC with matched liver and lung me-tastases, suggesting that the proportion of tumours with high immune cell infiltration together with PD-L1-positivity almost doubled in metastases compared to primary tumours. Moreover, differences observed in immune contexture between primary tumours and metastases may have significance for treatment strategies for patients with advanced CRC. The strong point of this study is that few publications have reported a comparison of immune microenvironment between CRC metastases and corresponding primary tumours. Another strength of this study is the evaluation of metastatic immune infiltration from whole tumour sections.

As discussed by the authors, the study presents some limitation:

1) the number of patients with resected metastases is relatively small

2) inclusion of only resectable patients may subject to selection bias

In my opinion these points are essential for the study.

The authors  should replace some tables with figures that can better represent the data and that make the work less arduous.

I would advise the authors to suggest a possible therapy for this type of patient.

Finally, the authors should insert the most recent references.

Author Response

Response to Reviewer 3 Comments

TITLE:   Immune contexture of MMR-proficient primary colorectal cancer and matched liver and lung metastases.

Correspondence: Dr. Maarit Ahtiainen (Department of Education and Research, Hospital Nova of Central Finland, 40620 Jyväskylä, Finland)

Comments to the Author

Summary of the major findings and impressions.

Point 1: The aim of the study was to evaluate immune microenvironment and its prognostic value in a series of mismatch proficient (pMMR) CRC with matched liver and lung me-tastases, suggesting that the proportion of tumours with high immune cell infiltration together with PD-L1-positivity almost doubled in metastases compared to primary tumours. Moreover, differences observed in immune contexture between primary tumours and metastases may have significance for treatment strategies for patients with advanced CRC. The strong point of this study is that few publications have reported a comparison of immune microenvironment between CRC metastases and corresponding primary tumours. Another strength of this study is the evaluation of metastatic immune infiltration from whole tumour sections.

As discussed by the authors, the study presents some limitation:

1) the number of patients with resected metastases is relatively small

2) inclusion of only resectable patients may subject to selection bias

In my opinion these points are essential for the study.

Response 1: We appreciate that the reviewer pays attention to the strengths of our study as well as agrees with us for the recognized limitations.

Point 2: The authors should replace some tables with figures that can better represent the data and that make the work less arduous.

Response 2: We agree with the reviewer that tables are too many and they are rather crowded. Several improvements have been made as follows:

1) The upper part of Table 2 (TILs densities) has been restructured as a box plot in Figure 2. The lower part of Table 2 including categorized variables has been moved into Supplementary Materials Table S1.

2) Table 4 is replaced with Figure 5 including Kaplan-Meier curves for the entire cohort (OS and DSS) (A) and Forest plots for subgroup results (B).

3) Table 5 is replaced with Figure 6 including Kaplan-Meier curves.

Point 3: I would advise the authors to suggest a possible therapy for this type of patient.

Response 3: We feel that we are not qualified to suggest any specific therapy based on our current study. Still, we have speculated the possible usability of immune therapies also in pMMR CRC patients in Discussion section. Our study shows that the proportion of tumours potentially responsive to immunotherapy is highly increased in metastases compared to primary tumours in pMMR CRC patients. A recent review (Zhang et al, 2020) nicely gathers immunotherapy trials for pMMR CRC. Therefore, we added a following sentence into Discussion section´s second last paragraph: However, combination strategies are apparently needed for improving treatment efficacy in pMMR CRC[46].

Point 4: Finally, the authors should insert the most recent references.

In addition to previously mentioned review, we inserted two recent references (both from Ottaiano et al, 2020) dealing with immune microenvironment and evolution of mutational landscape in CRC liver and lung metastases. The studies nicely prove that certain changes in cancer-related genes and immune cell infiltration relate to cancer progression. These findings give new perspectives for cancer diagnostics and therapeutic strategies. The following sentence was added into the third paragraph of Discussion section: Recently, specific changes in cancer-related genes and immune cell infiltration in CRC metastases have been related to metastatic evolution, which produce new perspectives for cancer diagnostics and therapeutic strategies [36,37].

Round 2

Reviewer 1 Report

all requested corrections and improvements have been made. nice work.